# DHA/EPA (Omega-3) and LA/GLA (Omega-6) as Bioactive Molecules in Neurodegenerative Diseases

**DOI:** 10.3390/ijms241310717

**Published:** 2023-06-27

**Authors:** Christina Kousparou, Maria Fyrilla, Anastasis Stephanou, Ioannis Patrikios

**Affiliations:** School of Medicine, European University Cyprus, 6 Diogenous Str., 2404 Nicosia, Cyprus; christina.kousparou@cytanet.com.cy (C.K.); mf182195@students.euc.ac.cy (M.F.); a.stephanou@euc.ac.cy (A.S.)

**Keywords:** EPA/DHA/LA/GLA, omega-3, omega-6, PUFA, polyunsaturated, Parkinson’s disease, Alzheimer’s disease, multiple sclerosis, Huntington’s disease, amyotrophic lateral sclerosis, clinical trials

## Abstract

Neurodegenerative diseases are characterized by neuroinflammation, neuronal depletion and oxidative stress. They coincide with subtle chronic or flaring inflammation, sometimes escalating with infiltrations of the immune system cells in the inflamed parts causing mild to severe or even lethal damage. Thus, neurodegenerative diseases show all features of autoimmune diseases. Prevalence of neurodegenerative diseases has dramatically increased in recent decades and unfortunately, the therapeutic efficacy and safety profile of available drugs is moderate. The beneficial effects of eicosapentaenoic acid (EPA) and docosahexaenoic acid (DHA) polyunsaturated fatty acids (omega-3 PUFAs) are nowadays highlighted by a plethora of studies. They play a role in suppression of inflammation, gene expression, cellular membrane fluidity/permeability, immune functionality and intracellular/exocellular signaling. The role of omega-6 polyunsaturated fatty acids, such as linoleic acid (LA), gamma linolenic acid (GLA), and arachidonic acid (AA), on neuroprotection is controversial, as some of these agents, specifically AA, are proinflammatory, whilst current data suggest that they may have neuroprotective properties as well. This review provides an overview of the existing recent clinical studies with respect to the role of omega-3 and omega-6 PUFAs as therapeutic agents in chronic, inflammatory, autoimmune neurodegenerative diseases as well as the dosages and the period used for testing.

## 1. Introduction

Neurodegenerative diseases continue to be on the rise and are characterized by the loss of functional neurons in the brain which further cause impairment and disability. They have been identified as a critical public health problem, as they negatively affect cognitive and physical health whose alterations impact everyday activities and functionality which, in turn, are the major determinants of quality of life. The incidence of these diseases, specifically Alzheimer’s and Parkinson’s diseases, rise dramatically with increasing age and the number of cases is reaching peak numbers as the lifespan is expanding [1]. 

Most neurodegenerative diseases are idiopathic and because of the elicited immune response and the self-antigens released, are classified as autoimmune diseases. The strongest predisposing factor is the genetic and genes involved in innate or adaptive immunity [2]. Multifactorial interactions are responsible for autoimmune diseases and an interplay between environment, lifestyle, microbes, exposure to toxins and nutrition play an essential role in the evolution of these diseases. Therefore, different environmental stimuli may activate the immune system and cause an aberrant immune response to self-antigens which might influence disease severity and symptom characteristics [3,4]. It has become increasingly clear that pathogenic events may occur years before clinical presentation, and when symptoms arise, it might be too late for a therapeutic opportunity. Precise determination of neurodegenerative disease remains a challenge and in up to 80% of cases the definite diagnosis is confirmed post-mortem [4].

Currently, there are no available curative treatments for any form of neurodegenerative disease, and despite many promising innovations in the field, treatment strategies fail to prevent neuronal loss. Disease-modifying treatments focus on alleviating symptoms and on slowing disease progression [5]. 

A plethora of studies have highlighted the anti-inflammatory actions of polyunsaturated fatty acids (PUFAs) and their potential use as therapeutic agents in neurodegenerative diseases, such as Parkinson’s disease, dementia, Alzheimer’s disease, multiple sclerosis, Huntington’s disease, and amyotrophic lateral sclerosis [6].

This review will provide an update on the role of docosahexaenoic acid (DHA)/eicosapentaenoic acid (EPA) (omega-3) and linoleic acid (LA)/gamma linolenic acid (GLA) (omega-6) molecules in the management of neurodegenerative diseases, and an evaluation of the current data based on clinical studies on their role as therapeutic agents.

## 2. The Leading Role of Lipids

Lipids are considered a heterogenous group of molecules that have hydrophobicity as a common property. Their structure ranges from simple short hydrocarbon chains to more complex chains, including triacylglycerols, sterols, sphingolipids, and phospholipids. Based on their length, degree of saturation, and hydroxylation, their biophysical properties are determined [7]. 

They are implicated in the regulation of the expression of transcription factors and several metabolic processes, such as fatty acid synthesis, oxidation, insulin sensitivity, and central nervous system function [8]. Fatty acids are divided into different categories that include saturated, monounsaturated, polyunsaturated, cis and trans fats [9].

Fatty acids in general are major components of all forms of lipids and, together with cholesterol and coalesce in the cell membrane, form the lipid bilayer of cells and organelles [10]. They are derived either de novo or from exogenous sources. In the human body, specific fatty acids can be synthesized from glucose or through metabolism of different lipid precursors. Different food sources contain different amounts and types of fatty acids that can be further esterified and/or metabolized into other forms of fatty acids or lipids. Cooking methods can affect the fatty acid content of various foods to a degree, of even becoming dangerous for health, due to the formation of polymers with hemagglutinin characteristics, saturated lipids, and fatty acids in trans stereochemical structure [11]. Different cells have different fatty acid composition that influences the membrane’s fluidity/permeability, as well as the function and movement of membranous proteins [12]. Membrane phospholipid fatty acids most frequently contain 12 to 24 carbon atoms forming hydrocarbon chains [13].

## 3. PUFAs Structure and Role

PUFAs are a type of fatty acids that contain two or more double bonds within their hydrocarbon chain. PUFAs can be classified based on the position of the initial double bond in relation to the omega methyl group located at the end. Omega-3 or omega-6 PUFAs are distinguished by the presence of a double bond positioned three or six atoms away from the omega terminus carbon, respectively. These PUFAs exhibit amphipathic characteristics due to the hydrophobic lipid tails and the hydrophilic phosphate-rich heads on the outer side [14,15]. 

Consumption of PUFAs results in their penetration and incorporation into the cell membranes from where they can exert actions on cell functions. More specifically, apart from maintaining cell membrane fluidity, they affect many functions of the cells including decreasing secretion of cytokines by monocytes, decreasing susceptibility to ventricular rhythm disorders, affecting specific cellular movement and translocation, and inhibiting platelet aggregation [16,17].

The human body can produce limited amounts of LA and a-linoleic acid (ALA) which are the precursor molecules for the production of any other form of PUFAs and therefore are called essential fatty acids. For that reason, external supplementation is needed to meet the demand. It has been clearly reported that essential fatty acid shortage can potentially contribute to dermatitis, renal hypertension, mitochondrial activity disorders, cardiovascular diseases, type 2 diabetes, impaired brain development, arthritis, depression, and decreased body resistance to infection [16,17].

Some of these defects may be due to low intake of omega-3/omega-6 PUFAs like alpha-linolenic acid (omega-3) and/or linoleic acid (omega-6). They might also be in relation to the metabolic products of LA and ALA, including the long-chain omega-6 PUFAs arachidonic acid (AA, 20:4 omega-6) and the long-chain omega-3 PUFAs eicosapentaenoic acid (EPA, 20:5 omega-3) and docosahexaenoic acid (DHA, 22:6 omega-3) [16].

All aforementioned essential molecules play an important role in mediating inflammatory responses and exert a wide spectrum of biologic activity in different body systems. The three major subtypes of eicosanoids and their major biologic actions are summarized in Figure 1.

As previously mentioned, inflammation is involved in neurodegenerative disorders and cognitive decline. The relationship between inflammation and oxidative stress is bidirectional: oxidative stress induces inflammation and inflammation induces oxidative stress (Figure 2). Hence, agents that act to reduce oxidative stress can also be considered as anti-inflammatory. 

Resolution of inflammation has always been viewed as a passive process, occurring because of the withdrawal of proinflammatory signals which further includes lipid mediators, such as leukotrienes and prostaglandins, as shown earlier [20]. Recently, it has been established that inflammation resolution is an active process with a distinct set of chemical mediators, including PUFAs. Molecules, such as resolvins and protectins, are nowadays identified as molecules that are generated from omega-3 PUFA precursors and can orchestrate the timely resolution of inflammation in model systems [20]. 

### 3.1. Omega-3 and Omega-6 PUFAs

α-linolenic acid (ALA, 18:3 omega-3), eicosapentaenoic acid (EPA, 20:5 omega-3) and docosahexaenoic acid (DHA, 22:6 omega-3) are all omega-3 fatty acids whilst linoleic acid (LA, 18:2 omega-6) and arachidonic acid (AA, 20:4 omega-6) belong to omega-6 fatty acids (see Figure 3). Although the human body cannot synthesize them, mainly because of the deficiency of one of the conversion enzymes, the omega-3-desaturase, it is able to metabolize them [21].

Through elongation stages, ALA is metabolized to EPA and DHA by two specific enzymes, Δ6 desaturase and Δ5 desaturase, whilst LA is metabolized to AA. EPA and ALA both compete for the same enzyme system, therefore, high background n-6 PUFA intake reduce interconversion of n-3 PUFAs (Figure 4).

As previously mentioned, AA is known as the precursor of proinflammatory mediators including prostaglandins and leukotrienes which promote inflammation [23]. DHA and AA are the most important PUFAs in the human brain with DHA being the major PUFA that has a prominent role in brain development. More specifically, more than 90% of the omega-3 PUFAs and 20% of the total brain lipids consists of DHA. DHA is incorporated in phosphatidylcholine, phosphatidylserine and at synaptic terminals and endoplasmic reticula. Among DHA’s actions, modulation of cellular properties, release of neurotransmitters and neuronal growth and gene expression are noted. Although several questions remain partially answered, this molecule is very promising and further research could identify a solid correlation between high DHA concentrations and neuroprotection [24].

### 3.2. PUFAs Transportation to the Brain

Research conducted in laboratory settings and living organisms has provided evidence that dietary intake of EPA, DHA, LA, and GLA can play a role in influencing and regulating various intricate networks of events and pathways involved in brain pathophysiology. The composition of fatty acids in the brain’s membranes can be altered through dietary supplementation, although this process has been observed to be influenced by age (taking longer time in adults compared to developing brains) and possibly influenced by the quantity of PUFAs consumed or supplemented. Both human and animal studies have demonstrated that diets rich in DHA and EPA can elevate the proportion of these PUFAs in the membranes of inflammatory cells while simultaneously decreasing the AA levels [16].

Oral supplementation increases the content of omega-3 PUFAs in the cerebrospinal fluid, although efficient passage through the blood–brain barrier requires a carrier particle. DHA needs 1-lyso, 2-docosahexaenoyl-glycerophosphocholine (LysoPC-DHA), which is brain specific to be transported to the brain. Carriers able to transport DHA to the brain with better properties are further studied and research is promising [25]. An example of a potential carrier with superior characteristics is AceDoPC (1-acetyl,2-docosahexaenoyl-glycerophosphocholine). This is a structured glycerophospholipid that facilitates the transport of DHA and has been shown to be associated with neuroprotective properties [26].

### 3.3. Omega-3 and Omega-6 Dietary Sources

As omega-3 and omega-6 are considered essential fatty acids, diet has a vital role in providing them [27]. There are various sources of long-chain PUFAs, both aquatic as well as animal. Oily fish of cold water is considered as an excellent source of long-chain omega-3 PUFAs, predominantly EPA and DHA. Fatty acids from animal sources, such as beef, lamb, pork, poultry, and dairy products are influenced by the diet and the digestive system of each animal. More specifically, muscle and adipose tissue of meat are rich in ALA, EPA, DPA and DHA [27]. DHA is found in high concentrations (0.7%) in egg yolk and its concentration can even increase when a chicken’s diet is supplemented with fish oils. Important plant sources of ALA include black raspberry seed oil which reaches a concentration of 35% as well as cranberry, basil seed oil, chia seed oil, walnut seed oil and flaxseed oil. LA is found in high concentrations in safflower and corn oils which can be further metabolized to other omega-6 fatty acids [27].

### 3.4. Ratio of Omega-3 to Omega-6 PUFAs

Several studies suggest that the ratio of omega-3/omega-6 should be 1:1, but in the industrial countries the ratio is about 1:20 due to the high quantities of omega-6 in the everyday diet, fast foods and especially AA. An unbalanced omega-3/omega-6 ratio compromises, among others, the brain’s cytoarchitecture and functioning, cellular integrity and the physiological status of the immune cells as well [28]. 

A diet high in AA has a negative impact on health as it promotes the pathogenesis of several diseases as previously mentioned. In contrast, it is supported that omega-3 PUFAs, are able to affect neuronal transmission by changing the phospholipid composition and can positively contribute on the fluidity of the central nervous system cellular membranes [12]. Therefore, the strong relationship between PUFAs’ status and brain functions, such as neurotransmission and behavior are highlighted, whilst a diet poor in n-3 PUFAs is considered detrimental to health [29].

There is accumulating scientific evidence on the possible efficacy of PUFAs supplementation in neurodegenerative disorders. Although dietary recommendations are far from being accepted as treatment for neurodegenerative disorders, they may be able to alleviate some of the symptoms and most importantly slow cognitive and physical decline which have the highest impact on the quality of life [30].

Omega-3 PUFAs are considered the most prescribed supplements and it is predicted that their use will rise by 6.5% from 2023 to 2032; their market segment is estimated to exceed 4.5 billion dollars by 2032 [31]. As omega-3 fatty acids are supplements that do not need testing and approval by official approval bodies, routine clinical practice in diseases should provide evidence for their role [30].

## 4. Parkinson’s Disease 

Parkinson’s disease (PD) is the second most prevalent neurodegenerative disease following Alzheimer’s disease and it affects 2–3% of the population older than 65 years old [32]. Neuropathological hallmarks of PD include neuronal degeneration in the substantia nigra pars compacta which contributes to striatal dopamine deficiency and intracellular inclusions containing aggregates of α-synuclein known as Lewy bodies [33,34]. Substantia nigra is a region in the brain responsible for movement control and clinical manifestations of PD, including bradykinesia, rigidity, tremor, and disturbed body balance [35]. Although environmental and genetic factors play a role in the pathogenesis of the disease, the underlying pathogenesis has not been fully elucidated [36]. Involvement of various pathways and mechanisms are thought to play a critical role among which mitochondrial dysfunction, oxidative stress, α-synuclein proteostasis and neuroinflammation are included. Treatment mainly focuses on substitution of striatal dopamine in addition to approaches to address motor and non-motor symptoms [37]. 

There are several scales commonly used to assess PD in clinical and research settings. These scales are designed to evaluate various aspects of PD symptoms, motor function, and quality of life [38]. Some of the commonly used scales for assessing PD are listed below in Table 1.

Many clinical trials, both prospective and observational, have studied the correlation of supplementation with PUFAs to the delay in progression and improvement of symptoms of Parkinson’s disease. 

The Honolulu-Asia Aging Study, 1965, was the first major prospective study that recorded environmental, lifestyle and physical attributes in a cohort of 8006 Japanese American men for 30 years. Among the dietary factors tested, PUFAs were included and showed an inverse association with PD [39].

In agreement to this study, the Rotterdam Study was a prospective population-based cohort study aimed at finding an association between the intake of unsaturated fatty acids and the risk of incident PD on previously healthy individuals. No associations were found between intake of PUFAs and Parkinson’s disease that could indicate a neuroprotective effect [40].

The Health Professionals Follow-up Study and the Nurses’ Health Study tried to evaluate the association between dietary lifestyle and the risk of PD. Two different dietary lifestyles were compared, one included a prudent diet which was characterized by high intake of fruits and fibers, and the other was a typical Western diet. The outcome of this study highlighted the importance of a healthy, balanced diet as patients who followed a prudent diet had a reduced risk of PD [41].

Additionally, a case–control study performed in Japan examined the relationship between intake of dietary fatty acids and the risk of PD in 249 cases. It was demonstrated that higher consumption of AA may be related to an increased risk of PD, whereas consumption of total fat, saturated fatty acids, monounsaturated fatty acids (MUFAs), omega-3 PUFAs, ALA, EPA, DHA, omega-6 PUFAs, and LA and a ratio of omega-3 to omega-6 polyunsaturated fatty acid intake were not [42]. 

Pantzaris et al. have studied the effects of a nutritional formula, Neuroaspis PLP10^®^, rich in omega-3 and omega-6 fatty acids and increased amount of gamma-tocopherol in PD [43]. Forty patients were enrolled into two groups, with one group receiving 20 mL of pure virgin olive oil and the other group receiving 20 mL Neuroaspis PLP10^®^, once daily; a mixture of omega-3 (810 mg EPA and 4140 mg DHA, I:5 *w*/*w*) and omega-6 fatty acids (1800 mg GLA and 3150 mg LA, 1:2 *w*/*w*) with overall omega-3 to omega-6, 1:1 *w*/*w*, including 0.6 mg vitamin A, 22 mg vitamin E and pure gamma (γ)-tocopherol (760 mg), for a total of 30 months in a randomized double-blind, placebo-controlled trial. Patients were assessed using the HY and UPDRS III scales. Neuroaspis PLP10^®^ was found to delay disease progression when used as an adjuvant to the formal therapy based on the UPDRS scale [43].

With regard to omega-3 PUFAs’ anti-inflammatory properties, Moralez da Silva et al., 2008, investigated the association between omega-3 fatty acid supplementation and depression in PD patients [44]. Patients were blindly separated into two groups: the first group received 480 mg/day DHA + 720 mg/day EPA from fish oil with vitamin E, whilst the second group received mineral oil, both groups versus placebo for three months. Furthermore, the group on the intervention was receiving antidepressant but not the group on placebo. PD patients on the intervention scheme showed decreased depression symptoms vs. the ones on placebo. However, larger sample size studies are required to confirm a possible correlation [44].

Lastly, Taghizadeh et al., 2017, studied the effects of omega-3 fatty acids and vitamin E supplementation on clinical and metabolic status of PD patients. Sixty PD patients participating in the study were randomized into two groups. One group was taking 1000 mg omega-3 fatty acids from flaxseed oil + 400 IU vitamin E while the other group was on placebo supplementation. After 12 weeks, omega-3 fatty acids and vitamin-E supplementation led to a statistically significant improvement in UPDRS (*p* = 0.02) [45].

Unfortunately, randomized clinical trials for PD based on omega fatty acids-based formulation are few because of poor patient adherence to therapy due to palatability and taste issues of the formulas, long duration of the study and the time-consuming evaluation process. A period of three to six months is shown to be not enough for a supplement to act, and especially the omega fatty acid-based supplements, therefore, longer duration studies are needed to establish a strong correlation [30].

## 5. Multiple Sclerosis 

Multiple sclerosis (MS) is a multifactorial, chronic neurodegenerative autoimmune disease of the central nervous system and spinal cord which causes axonal damage. The incidence of the disease is rising, affecting more than 2 million people worldwide, and the socioeconomic impact is detrimental. The mechanism of cause behind MS is still opaque, but it has been shown that it involves immune-mediated inflammation, oxidative stress and excitotoxicity, all of which contribute to oligodendrocyte and neuronal damage and even cell death, hence promoting disease progression [46,47].

Many environmental factors have been identified as causatives, such as modern/industrial countries’ dietary habits, low vitamin D, exposure to specific infections, smoking, and obesity among others. Clinical aspects include loss of vision, paraplegia and spasticity in various parts of the body [48,49]. Diagnosis is definite when an MRI is performed which indicates inflammatory lesions and axonal loss. The McDonald criteria are used to guide physicians in establishing MS diagnosis and MRI. Its increasing prevalence and minimal treatment efficacy and side effects have made the development of new treatments imperative [50].

It has been long suggested that PUFAs deficiencies might play a role in the pathogenesis of the disease. Increased inflammation and increased amounts of AA promote the formation of proinflammatory eicosanoids, as previously discussed, alter glutamates homeostasis, causing oligodendroglial death. Many studies have identified the crucial role of dietary EPA, DHA, LA and GLA and their possible association with MS progression [51]. 

Pantzaris et al., 2013, studied the effects of a novel oral nutraceutical formula of omega-3, omega-6 and specific vitamins (PLP10) in relapsing remitting (RR) MS. A randomized double-blind, placebo-controlled clinical trial was conducted on 80 MS patients. Participants were randomly allocated into four groups. The first group received omega-3 and omega-6 PUFAs at a 1:1 *w/w* ratio. Specifically, the omega-3 fatty acids were DHA and EPA at a 3:1 *w/w* ratio, and the omega-6 fatty acids were LA and GLA at 2:1 *w/w* ratio. This intervention also included minor quantities of other specific polyunsaturated, monounsaturated and saturated fatty acids as well as vitamin A and vitamin E (α-tocopherol). The second group received PLP10 a formula containing all the above plus γ-tocopherol. The third group received γ-tocopherol alone, whilst the fourth group of participants received pure virgin olive oil as placebo. The interventions were administered via mouth once daily, 30 min before dinner for 30 months. The main outcome measured was the annual relapse rate (ARR) for each one of the four groups for a time period of 2 years on treatment. The results suggested that patients receiving PLP10 had reduced ARR and disability progression as measured by the expanded disability status scale (EDSS) [28]. 

In 2022, the same research group published the findings from a 30-month phase III multicenter study that was randomized, double-blind, and placebo-controlled. This trial involved 61 patients diagnosed with RRMS who were currently receiving IFN-β treatment [52]. The participants were randomly divided into two groups: one received Neuroaspis PLP10^®^ (*n* = 32), while the other received a placebo (*n* = 29). Over the course of 30 months, the patients orally consumed 20 mL of Neuroaspis PLP10^®^ or placebo once daily. The primary outcome measured was the ARR, while secondary outcomes included sustained disability progression assessed through the Expanded Disability Status Scale (EDSS), as well as brain T2 and gadolinium-enhancing lesions after 2 years. The trial demonstrated that Neuroaspis PLP10^®^ led to a significant 80% reduction in ARR (relative risk reduction (RRR): 0.20; 95% confidence interval (CI): 0.09 to 0.45; and *p* = 0.0001) compared to the placebo group. Furthermore, there was a 73% decrease in the risk of sustained disability progression (hazard ratio (HR): 0.27; 95% CI: 0.09 to 0.83; and *p* = 0.022) at the 2-year mark. The number of T1 gadolinium-enhancing lesions and the number of new/enlarged T2-hyperintense lesions were also significantly reduced (*p* = 0.01 and *p* < 0.0001, respectively). Both T1-enhancing and new/enlarging T2-hyperintense lesions showed significant reduction (*p* = 0.05 and *p* < 0.0001, respectively). No notable adverse events were reported. Consequently, the study concluded that the addition of Neuroaspis PLP10^®^ to IFN-β treatment significantly proved to be more effective than IFN-β treatment alone in patients diagnosed with RRMS [52].

Shinto et al. in 2016, studied the association between omega-3 supplementation and treatment-resistance major depressive disorder (MDD) in 39 patients with MS in a randomized, double-blind placebo-controlled pilot study. The product under investigation was a fish oil capsule that contained both EPA and DHA with an EPA/DHA ratio of 1.4:1.0 in a triglyceride form. Each gel capsule contained 0.968 g of fish oil with 0.325 g EPA and 0.225 g DHA and 0.640 g of total omega-3 fatty acids. Participants randomized to the omega-3 FA group received a daily dose of six capsules (1.95 g of EPA and 1.35 g of DHA). Participants randomized to the placebo group received capsules that contained soybean oil with 1% fish oil to taste and smell like the fish oil capsules. The Montgomery–Asberg Depression Rating scale (MADRS) was used to assess the results. Omega-3 fatty acids supplemented patients did not show any improvement and no association between omega-3 and reduction in MDD was observed [53].

Additionally, the OFAMS study was a randomized, double-blind, placebo-controlled trial that was conducted from 2004 to 2008 which investigated whether omega-3 supplementation in combination with beta-1a-treatment reduces MRI findings of MS as well the patients’ clinical picture. More specifically, patients received 1350 mg of EPA and 850 mg of DHA daily or placebo. After 6 months, all patients received additionally subcutaneously 44 μg of interferon beta-1a, three (3) times per week for another 18 months. Unfortunately, results showed that omega-3 supplementation did not have an effect on disease activity compared to the placebo group [54]. 

AlAmmar et al., 2021, reported that omega-3 supplementation contributes to decreased levels of proinflammatory cytokines and free radicals as well as improved quality of life in MS patients. They screened thousands of studies, and the seven which met their inclusion criteria (namely studies performed on humans both male and female, aged 18 years at minimum, diagnosed with MS according to the McDonald 2010 criteria), showed the beneficial roles of fish oil supplementation and omega-3 fatty acids in improving the quality of life of MS patients. These roles were attributed to their beneficial effects on inflammatory markers, glutathione reductase, reducing the relapsing rate, and achieving balanced omega-6 to omega-3 ratios [55].

The Ausimmune study took place in Australia between 2003 and 2006 [56]. In this case–control study, the intake of dietary fat in relation to risk of a first clinical diagnosis of CNS demyelination (FCD) was examined. Dietary data were collected using a validated food frequency questionnaire. In 267 cases and 517 controls with dietary data, higher intake (per g/day) of omega-3 PUFA, particularly originated from fish rather than from plants, was associated with a decreased risk of FCD. Total fat intake and intake of other types of fat were not associated with FCD risk. It was concluded that participants that were consuming PUFAs had a significantly decreased risk of CNS demyelination [56].

In multiple sclerosis (MS), MMP-9 is believed to play a crucial role in facilitating the migration of inflammatory cells into the central nervous system (CNS) by contributing to the disruption of the blood–brain barrier [57]. On the other hand, omega-3 fatty acids have been observed to decrease the levels of proinflammatory cytokines and also act by reducing the levels of matrix metalloproteinases (MMPs). In a relevant open-label study, a group of ten individuals with relapsing–remitting MS (RRMS) were provided with omega-3 fatty acid supplementation (9.6 g/day of fish oil). The participants were assessed at four different time points: baseline, after one month of omega-3 supplementation, after three months of omega-3 supplementation, and following a three-month washout period. The study found that the secretion of MMP-9 by immune cells decreased by 58% after three months of omega-3 fatty acid supplementation compared to baseline levels (*p* < 0.01) and by 45% after the 3-month of wash out (*p* = 0.01). Additionally, there was a significant increase in the levels of omega-3 fatty acids in the membranes of red blood cells, indicating their potential as immune-modulators with therapeutic benefits for patients with MS [57].

While more research is needed to further confirm the benefits of omega-3 fatty acids for MS, these initial findings suggest that PUFA supplementation results in a decrease in proinflammatory cytokines, free radicals, and as a result, improves the quality of life of patients with MS by decreasing their relapse rates [58,59]. 

## 6. Huntington’s Disease

Huntington’s disease (HD) is a rare and late-onset neurodegenerative disorder that affects the central nervous system, and it is characterized by degeneration of the striatum and more specifically of the caudate nucleus and putamen and loss of efferent medium spiny neurons. HD is most prevalent in the United States of America and in the United Kingdom and also prevalent in countries, such as Japan, Greece, and Sub-Saharan Africa [60]. The mean age at diagnosis is 30–50 years old and its clinical characteristics include unwanted chorea that gradually spreads to all the muscles from distal to more proximal ones, cognitive decline, and psychiatric disturbances [61]. Additional features include weight loss, cachexia, and heart failure. Cachexia is associated with skeletal muscle wasting and progressive loss of motor function, weakness, and fatigue [62]. HD is considered autosomal dominant, and it is caused by a CAG elongation on the short arm of chromosome 4 within the huntingtin gene [63]. An inverse correlation between the number of CAG repeats and age of onset is known and, as the disease is passed to the next generations, it appears much earlier in life [64]. This phenomenon is called anticipation and can be explained by the increase in size of CAG repeats that are passed through the male germline [65]. Furthermore, it has been reported that monozygotic twins have different clinical symptoms and epigenetic factors and variations in CAG repeats are mostly responsible [66]. No cure for HD is known and treatments focus on symptomatic relief and quality of life and not disease progression [67]. 

Indirect evidence suggests that omega-3 PUFAs help to alleviate symptoms related to HD. More specifically they help to reduce cachexia and weight loss as well as decrease cognitive decline over time. Although evidence is still not strong, diet could affect HD onset and expression [60]. For instance, it has been proposed that a tight correlation between omega-3/omega-6 ratio and HD prevalence exists. Western diet is composed of a high omega-6/omega-3 (16.7:1 *w*/*w*) ratio which is linked to the disease’s high incidence in the US. Furthermore, omega-6 PUFAs are found to be involved in HD pathogenesis and progression [68]. As previously stated, a high omega-6/omega-3 ratio is linked to inflammation, osteoporosis, and cancer [69].

Smith et al., 2015, investigated whether PUFAs were able to slow muscle loss and function. Sixty healthy individuals between 60 and 85 were randomly assigned in a 2:1 fashion to either omega-3 PUFA therapy [four 1 g pills/day omega-3 acid ethyl esters that provided a total of 1.86 g EPA (20:5 omega-3) and 1.50 g DHA (22:6 omega-3)/day, which is equivalent to the omega-3 PUFA content of 200–400 g freshwater fatty fish (e.g., salmon, herring, and sardines)] or a placebo control (four identical looking pills/day with corn oil) for 6 months. Thigh muscle volume, handgrip strength, one-repetition maximum (1-RM) lower- and upper-body strength, and average power during isokinetic leg exercises were evaluated before and after treatment. It has been shown that omega-3 PUFAs were able to slow muscle loss, and the study was considered promising as it suggested a positive correlation between muscle loss and omega-3 supplementation but was not extrapolated to the weight loss seen in HD [70].

Unfortunately, inadequate research has been performed to examine the effects of omega-3 PUFAs, specifically in HD, and their possible therapeutic use on symptom control and disease progression. Various studies’ results remain controversial as, while some find a correlation, other do not.

## 7. Dementia and Depression

The anti-inflammatory effects of PUFAs can lower the risk of neurological and psychiatric symptoms as well as help to stabilize the mood. They have been associated with high cognitive performance and decreased incidence of dementia [71].

Nooyens et al., 2018, studied cognitive function in 2612 individuals participating in the Doetinchem Cohort Study to test the association between fish and different fat intakes as well as a 5-year change in cognitive functions [72]. Fish consumption was evaluated in terms of overall frequency and relative frequencies within three distinct fish groups: lean or moderately fatty fish (such as flounder, and codfish), shellfish (such as mussels, and shrimps), and fatty fish (such as salmon, herring, and eel). Fat intake, including total fat, saturated fat, MUFAs, PUFAs, and cholesterol, was calculated using the Dutch food composition table from 1996 based on the responses obtained from a food frequency questionnaire. The intake of omega-3 PUFAs (ALA, DHA, and EPA) and the primary omega-6 fatty acid (LA) was calculated for all food items using the 2001 version of the Dutch food composition table since the 1996 version did not provide the omega-3 PUFA contents (ALA, EPA, and DHA) at that time. The study analyzed these intakes in relation to changes in global cognitive function, memory, information processing speed, and cognitive flexibility over a period of five years. No consistent link was found between (fatty) fish consumption and cognitive decline. However, a higher intake of omega-3 PUFAs, particularly ALA, was associated with a slower decline in global cognitive function and memory (*p* < 0.01). The consumption of other fatty acids did not exhibit any significant associations with cognitive decline [72].

Cutuli et al., 2017, have highlighted that omega-3 PUFAs may have a beneficial role in neuroplasticity and cognitive impairment. In their review, they primarily examined the neuroprotection exerted by omega-3 PUFAs on cognitive impairment and markers of reduced brain plasticity and neurodegeneration during non-pathological aging. Multiple pathways of omega-3 PUFA preventive action were considered. Omega-3 PUFAs have been shown to increase the levels of several signaling factors involved in synaptic functions, thus leading to the increase in dendritic spines and synapses as well as to the enhancement of hippocampal neurogenesis even at old age but it might take longer [73]. 

Thesing et al., 2018, examined the cross-sectional association of omega-3 and omega-6 PUFA levels (both in absolute values and their ratios with total fatty acids) in 2912 participants with a mean age of 41.9 years suffering from remitted or current depressive and anxiety disorders. Compared to controls, current comorbid depressive and anxiety disorder patients had lower omega-3 PUFA levels (*p* = 0.012), and lower omega-3/fatty acid ratios (*p* = 0.002) as did current pure depressive disorder patients (*p* = 0.021), whereas omega-6 PUFA levels were not different. No differences in PUFA levels were found between remitted patients and controls. Within patients, lower omega-3 PUFA levels were only associated with higher depression severity (*p* = 0.023), whereas for omega-6 PUFA levels and other clinical characteristics no clear association was observed. PUFA alterations were not associated with pure anxiety [74].

Grosso et al., 2016, performed a meta-analysis of results from observational studies exploring the association between fish, omega-3 PUFAs dietary intake, and depression. A total of 31 studies, including 255,076 individuals and over 20,000 cases of depression, were examined. Analysis of 21 datasets investigating the relation between fish consumption and depression resulted in significant reduced risk (RR = 0.78, and 95% CI: 0.69, 0.89), with a linear dose–response despite with moderate heterogeneity. Pooled risk estimates of depression for extreme categories of both total omega-3 PUFA and fish-derived omega-3 PUFA (EPA + DHA) resulted in decreased risk for the highest compared with the lowest intake (RR = 0.78, and 95% CI: 0.67, 0.92 and RR = 0.82, and 95% CI: 0.73, 0.92, respectively) and dose–response analysis revealed a J-shaped association with a peak decreased risk for 1.8 g/d intake of omega-3 PUFA (RR = 0.30, and 95% CI: 0.09, 0.98) [75].

## 8. Amyotrophic Lateral Sclerosis

Amyotrophic lateral sclerosis (ALS) is a fatal neurodegenerative disease characterized by the degeneration of motor neurons [76]. This disease was first described by Jean-Martin Charcot in 1869. Although the disease has been detected all over the world, the Western Pacific area has a 50–100 times higher prevalence [77]. 

Most ALS cases are sporadic (SALS). Approximately 5–10% of cases are familial form of the disease (FALS), in which 20% have a SOD1 gene mutation and approximately 2–5% have mutations in the TARDBP gene (TAR DNA binding-protein, TDP-43) [78]. Additionally, mutations in this gene also occur in SALS. ALS cases peak in adulthood and the average survival from symptom onset is 3–5 years, although 20% of patients may survive longer, up to 10 years from the initial symptom onset [79]. Males have a higher incidence of getting the disease in comparison to females and the overall population-based risk for men is 1:350 and for women is 1:400 [80]. However, recent research suggests that ALS affects both genders equally. The exact cause of the disease and specific mechanism of neuronal death is currently unknown. Increasing evidence suggests that an underlying autoimmune mechanism contributes to ALS pathogenesis and most specifically T-lymphocytic infiltration and macrophages in the anterior horn of the spinal cord [81].

Clinical hallmarks of the disease include a combination of upper and lower motor neuron symptoms with involvement of the brainstem and spinal cord. Most patients are diagnosed with the spinal form of the disease and the symptoms range from focal muscle wasting that gradually progresses to spasticity and weakness that later leads to atrophy. Bulbar form is less frequently seen in the initial stages of the disease and manifests with swallowing difficulties and speech disturbances. Later in the course, it progresses to the limbic form. Limbic onset is seen in two-thirds of the patients but later in the course of the disease they develop bulbar signs as well [82]. 

Research suggests that more than half of the ALS patients may have concomitant frontotemporal dementia which further complicates the situation. Atypical symptoms may be present, such as weight loss, emotional instability, muscle cramps and fasciculations [80,82]. Unfortunately, no therapy is available to cure this fatal disease and symptom control remains the focus of treatment. Extensive research and trials need to be performed in order to better understand the pathophysiology of the disease and provide new treatments [80].

A meta-analysis conducted by Fitzgerald et al. has tried to examine the association between omega-3 and omega-6 PUFAs consumption and ALS risk. Analysis was based on 1,002,082 participants (479,114 women; 522,968 men) in five prospective cohorts: the National Institutes of Health-AARP Diet and Health Study, the Cancer Prevention Study II Nutrition Cohort, the Health Professionals Follow-up Study, the Multiethnic Cohort Study, and the Nurses’ Health Study. Diet was assessed via a food frequency questionnaire developed or modified for each cohort. Participants were categorized into cohort-specific quintiles of intake of energy-adjusted dietary variables. A total of 995 ALS cases were documented during the follow-up. A greater omega-3 PUFA intake was associated with a reduced risk of ALS; the pooled, multivariable-adjusted risk ratio (RR) for the highest to the lowest quintile was 0.66 (95% CI: 0.53–0.81; and *p* trend < 0.001). Consumption of both ALA (RR = 0.73; 95% CI: 0.59 to 0.89; and *p* trend = 0.003) and marine omega-3 PUFAs (RR = 0.84; 95% CI: 0.65–1.08 and *p* trend = 0.03) contributed to this inverse association. Intakes of omega-6 PUFA were not associated with ALS risk [83].

We have included some animal-based studies as there is insufficient data in the literature on PUFAs, ALS and studies performed in humans. The following studies involved animal models. De Aguilar JL et al., 2004, reported that a high-fat diet (consisting of regular chow supplemented with 21% (*w*/*w*) butter fat and 0.15% (*w*/*w*) cholesterol) in transgenic mice was able to reverse muscle denervation and delay motor neuron death [84]. Yip et al., 2013, have studied the hypothesis that high EPA levels could be beneficial in ALS. A well-established mouse model of ALS with SOD1 mutations was used and exposed to dietary EPA (300 mg/kg/day). It was found that dietary EPA initiated at the disease onset had no effect on disease progression, whereas EPA supplementation at the pre-symptomatic stage shortened the lifespan [85]. Torres et al., 2020, studied the effect of DHA supplementation in G93A-SOD1 ALS mice. In this study, they evaluated the neuroprotective effect of dietary DHA supplementation in a preclinical model of motor neuron demise, the B6SJL-Tg (SOD1-G93A)1Gur/J transgenic mice. They investigated whether DHA nutritional interventions modulated omega-3 amounts in the spinal cord in the disease evolution, influencing survival, weight loss, motor function-related variables, and fatty acid profiles at three different ages (pre-symptomatic stage, disease onset, and endpoint). They concluded that male mice had an increased survival and slower ALS progression in comparison to female mice. This has been explained by estrogen-sensitive differences of females in microglia [86]. 

Unfortunately, no clinical investigations performed on humans were found in the literature concerning omega-3 and omega-6 effects on ALS. Studies performed on mice indicated a positive association between omega-3 and ALS risk reduction [84,85,86]. Therefore, the need to conduct human investigation, specifically on ALS using omega-3/omega-6 fatty, acids is of high importance especially since animal model studies reported well encouraging data.

## 9. Alzheimer’s Disease

Alzheimer’s disease (AD) is considered the most common form of dementia affecting approximately 62% of dementia patients and its prevalence is doubling every 5 years after the age of 65 [87]. It affects approximately 24 million people worldwide and is considered a major public health concern [88]. While most AD cases are sporadic, mutations in three specific genes are nowadays known. These are amyloid precursor protein (APP), presenilin 1 (PSEN1) and presenilin 2 (PSEN2). Typically, late-onset AD is associated with a combination of genetic and environmental factors. Nowadays, it is believed that more than 70% of AD risk is associated with genetic factors and more specifically with the APOE gene [89]. Genome association studies have identified more than 20 genetic risk factors including defects in inflammatory, cholesterol and endosomal recycling pathways. The combination of these risk factors can double one’s risk of developing AD [90].

AD is characterized by amyloid plaques and protein accumulation in the brain and Braak and Braak have mapped their movement and made associations between their movement and progression of the disease [91]. These proteins contribute to the loss of connection between nerve cells which leads to brain tissue death. Cognitive decline combined with memory loss, and difficulty in remembering names and details of known events, are the main characteristics of AD [92]. The mainstay of diagnosis focuses on clinical assessment and more specifically on the clinical interview and cognitive testing [93]. Although there are treatments available to fight this neurodegenerative disease, their focus is to alleviate symptoms as there are still no disease-modifying medications. 

The role of diet and more specifically omega-3 fatty acids is nowadays highlighted, and many studies have highlighted omega-3’s beneficial role in reducing comorbidity and improving patient’s clinical condition.

Lin et al., 2022, studied the role of omega-3 and blood biomarkers in AD and cognitive impairment in a multisite, randomized, double-blind, placebo-controlled trial [94]. A total of 163 mild cognitive impairment (MCI) or AD patients were randomly assigned to placebo (*n* = 40), DHA, 0.7 g/day, *n* = 41, EPA, 1.6 g/day, *n* = 40, or EPA (0.8 g/day) + DHA (0.35 g/day) (*n* = 42) group for 24 months. The results were measured as the cognitive and functional abilities, biochemical, and inflammatory cytokines profiles. They concluded that omega-3 PUFAS did not affect cognitive decline and depression, but they had a positive impact on language ability [94].

In 2015, Wu et al. conducted a meta-analysis of prospective cohort studies to examine the relationship between the intake of omega-3 fatty acids and the incidence of dementia and Alzheimer’s disease (AD). The researchers concluded that a higher consumption of fish was associated with a reduced risk of AD. However, they did not find statistical evidence supporting a similar inverse association between the intake of long-chain omega-3 fatty acids and the risk of dementia or AD. Additionally, the analysis did not demonstrate an inverse association between fish intake and the risk of dementia [95].

In 2021, Tofiq et al. organized the omega AD study which was a randomized controlled trial to investigate the effect of perioral omega-3 fatty acid supplementation on biomarkers in the cerebrospinal fluids of AD patients. Thirty-three patients were randomized to either treatment with a daily intake of 2.3 g of omega-3 FAs (*n* = 18) (four 1 g capsules daily, each containing 0.43 g DHA and 0.15 g EPA) or placebo (four isocaloric 1 g capsules daily, containing 1 g corn oil including 0.6 g linoleic acid) (*n* = 15). Cerebrospinal fluid (CSF) samples were collected at baseline and after six months of treatment, and several biomarkers were analyzed, but no significant differences between the groups were identified. Within the treatment group, there was a small but significant increase in chitinase-3-like protein 1 (YKL-40) (*p* = 0.04) and neurofilament light (NfL) (*p* = 0.03), indicating a possible increase in inflammatory response and axonal damage. This increase in biomarkers did not correlate with the Mini-Mental State Examination (MMSE) score [96].

In 2021, Zhu et al. conducted a prospective cohort meta-analysis aiming to investigate the impact of dietary fatty acids on Alzheimer’s disease (AD) and cognitive decline. The analysis included 14 studies, involving a total of 54,177 participants, including 1696 AD patients, 1118 dementia patients, and 2889 individuals with mild cognitive impairment (MCI). The combined relative risk (RR) revealed a significant association only between the intake of omega-3 PUFAs and the risk of MCI (RR, 0.86; and 95% confidence interval (CI), 0.75–0.98), with no substantial heterogeneity observed across the studies. However, the intake of total fatty acids, saturated fatty acids (SFAs), cholesterol, MUFAs, PUFAs, omega-3 PUFAs, omega-6 PUFAs, docosahexaenoic acid (DHA), and eicosapentaenoic acid (EPA) did not show a significant association with the risk of AD. Similarly, the intake of total fatty acids, SFAs, MUFAs, PUFAs, and omega-3 PUFAs was not significantly associated with the risk of dementia. Overall, this meta-analysis provided evidence suggesting that higher intake of omega-3 PUFAs may be linked to a reduced risk of MCI [97]. 

In 2016, Külzow et al. conducted a double-blind placebo-controlled proof-of-concept study to investigate the effects of omega-3 PUFAs on memory function. The study involved 44 cognitively healthy individuals aged 50–75 years, including 20 females. Participants were randomly assigned to receive either long-chain polyunsaturated omega-3 fatty acids at a dose of 2200 mg per day (*n* = 22) or a placebo (*n* = 22) for a duration of 26 weeks. The primary assessment focused on memory performance using the Object Location Memory (OLM) task, while secondary outcome measures included performance in the Rey Auditory Verbal Learning Test (AVLT), dietary habits, omega-3 index, and other blood-derived parameters. The results showed a significant increase in the omega-3 index among the participants who received the omega-3 supplementation compared to the placebo group. Additionally, the group that received omega-3 supplementation demonstrated significantly improved recall of object locations compared to the placebo group. However, there was no significant effect of omega-3 supplementation on performance in the AVLT. This study provided experimental evidence that omega-3 PUFAs have positive effects on memory functions in healthy older adults. These findings suggest the potential of novel strategies involving omega-3 supplementation to help maintain cognitive functions as individuals age [98].

In 2017, Canhada et al. conducted a systematic review with the aim of examining the effects of omega-3 fatty acids supplementation on Alzheimer’s disease (AD) [99]. The review identified seven studies that fully met the inclusion criteria. The majority of these studies did not report statistically significant results in favor of omega-3 fatty acids supplementation compared to a placebo. However, a few studies did show some benefits, but only in specific cognitive assessment scales. Interestingly, the effects of omega-3 fatty acids seemed to be more pronounced in patients with very mild AD. The researchers concluded that supplementation with omega-3 fatty acids may be beneficial in the early stages of AD when there is only slight impairment of brain function [99]. They emphasized the importance of further research in this area to establish a solid correlation between omega-3 fatty acids and AD.

In 2021, Soininen et al. reported the results of a 36-month multinutrient clinical trial called LipiDiDiet in prodromal AD. The trial investigated the effects of the specific multinutrient combination, Fortasyn Connect, which included among others DHA, and EPA, on cognition and related measures in prodromal AD. The study presented evidence suggesting that long-term multinutrient intervention in prodromal AD may lead to altered disease trajectories and positive effects. The intervention demonstrated significant benefits in terms of cognition, function, and brain atrophy, with clinically relevant effect sizes observed. Notably, the prolonged intervention period resulted in a wider range of statistically significant differences across various endpoints, surpassing previous reports. This study was the first to report sustainable benefits lasting for 3 or more years in an intervention targeting prodromal AD. Overall, the results emphasized that early and long-term intervention may enhance the benefits associated with this approach [100].

## 10. Discussion and Conclusions

It is now clear that there are a lot of controversial results and contradictory clinical study outcomes in relation to the beneficial role of the specific structural molecules highlighted as omega-3 (EPA and DHA) and omega-6 (LA and GLA). Searching the literature, several original as well as review studies reported that omega-3 PUFAs, specifically DHA and EPA, have a protective effect against neural degeneration. Meta-analyses of randomized controlled trials demonstrated that omega-3 supplementation significantly improves cognitive performance in older adults and those with mild cognitive impairment. Other studies have suggested that omega-3 PUFAs may have a role in improving synaptic function. But there are also studies reporting no effect or even worsening results. 

In addition to omega-3 PUFAs, omega-6 PUFAs, specifically AA, has a crucial role in neurodegenerative diseases. While AA is essential for normal brain function, excessive AA can lead to neuroinflammation and neuronal damage. Several studies have suggested that a high ratio of omega-6 to omega-3 PUFAs in the diet may increase the risk of developing neurodegenerative diseases; it has been linked to increased and uncontrolled inflammation which plays a key role in the disease development.

The debate on the optimal ratios of omega-3 to omega-6 is ongoing, as well as the ratio between the individual fatty acids, EPA to DHA and LA to GLA. It is suggested that most probably the ratio of 1:1 (omega-3 to omega-6) is likely more balanced.

The inconsistent findings regarding the significant efficacy of omega-3 and omega-6 fatty acids may be due in several important parameters with the most important ones being the absence of proper study protocols with solid inclusion and exclusion criteria and according to the international trial guidelines, absence of a run-in period prior to the entry baseline, variations in study populations, very short study periods, differences in supplement dosages used among studies, differences in the actual content of the individual molecules found in the intervention used in each trial, the use of fish extracts with many other impurities present in the samples and different ratio of active molecules, and with most studies using EPA in excess to the DHA. Moreover, different types of molecules were used as carriers, with some using triglycerides, other esterified fatty acids and with most studies referring to the use of omega-3 and omega-6 without indicating quantities, type, form of molecules or method of supply. On the other hand, there are studies reporting data out of solid protocols and well-performed clinical trials with the use of a well-defined cocktail formulation indicative of promising bioactivity associated with the aforementioned molecules when properly formulated and used. 

In conclusion, neurodegenerative conditions are considered a major challenging issue in clinical practice and, with the increase in lifespan, their burden is expected to increase in the forthcoming years. With no cure available, medical therapy remains symptomatic, but not curative. Literature is growing which highlights the relationship between omega-3 PUFAs, neuroprotection, and cell regeneration, and it represents an interesting biological potential. The safety associated with these molecules versus the side effects of conventional medicines are the parameters that drive nowadays the pharmaceutical industries to show even more interest on this multimillion dollars market pie. 

The integration of metabolomics with individualized therapies provides a promising approach for assessing the metabolic profiles and personalized treatment strategies involving omega-3 PUFAs to target the underlying mechanisms and improve outcomes in individuals. By analyzing the comprehensive profile of metabolites in biological samples, metabolomics allows for the identification and quantification of specific fatty acid metabolites, such as lipid species and their derivatives. This approach enables the assessment of metabolic alterations associated with neurodegeneration and the effects of fatty acid interventions. Metabolomic studies have revealed changes in lipid metabolism, oxidative stress markers, and inflammatory mediators in neurodegenerative disorders, providing a deeper understanding of the underlying pathophysiology. Additionally, metabolomics can help identify potential biomarkers for disease diagnosis, progression, and response to fatty acid interventions. By integrating metabolomics data with clinical and molecular information, personalized therapeutic strategies can be developed to optimize the use of fatty acids as a therapeutic approach in neurodegenerative disorders.

Adaptation of the diet and inclusion of PUFA-rich foods and supplements is highly recommended by the scientific community. Well-designed clinical trials are essential for advancing our understanding of the role of fatty acids in neurodegenerative diseases and for developing evidence-based recommendations for the use of fatty acid supplementation as a therapeutic intervention. To date, Neuroaspis PLP10^®^ nutritional supplement formula might be the only and most trialed intervention that is governed by solid protocol/well-defined studies, solid formulation and with a long on-trial follow-up period. On the other hand, the use of data out of those small sample sized trials might be considered a limitation; but the use of long trial periods strengthens the reliability of the outcomes and conclusions. Further studies based on properly designed protocols and proper formulation using the updated knowledge with larger sample population trials should be undertaken by pharmaceutical industries to concretely establish PUFAs as therapeutic agents for neurodegenerative diseases as well as other chronic diseases including cardiovascular. 

## Figures and Tables

**Figure 1 ijms-24-10717-f001:**
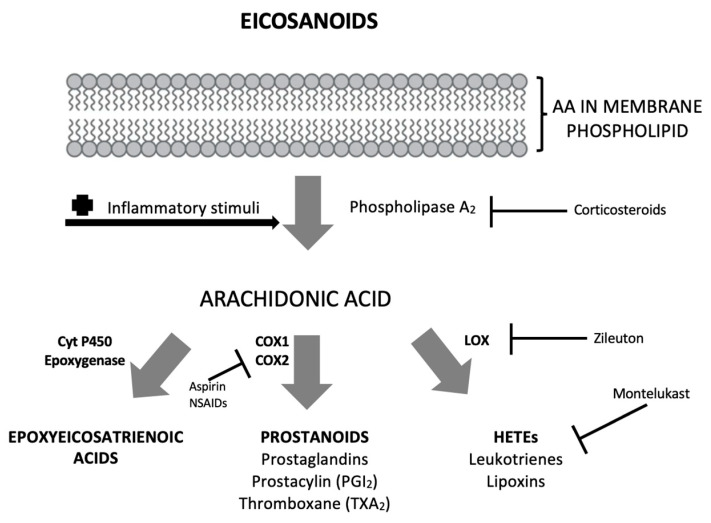
Pathway of biosynthesis of eicosanoids from arachidonic acid. Eicosanoids are not stored within cells and are synthesized as needed when their biosynthesis is activated by trauma/inflammation or cytokines which activate phospholipase A2 (PLA2). Fatty acids that are cleaved by PLA2 from cell membranes are then oxygenated by one of three different families of enzymes to produce eicosanoids [18].

**Figure 2 ijms-24-10717-f002:**
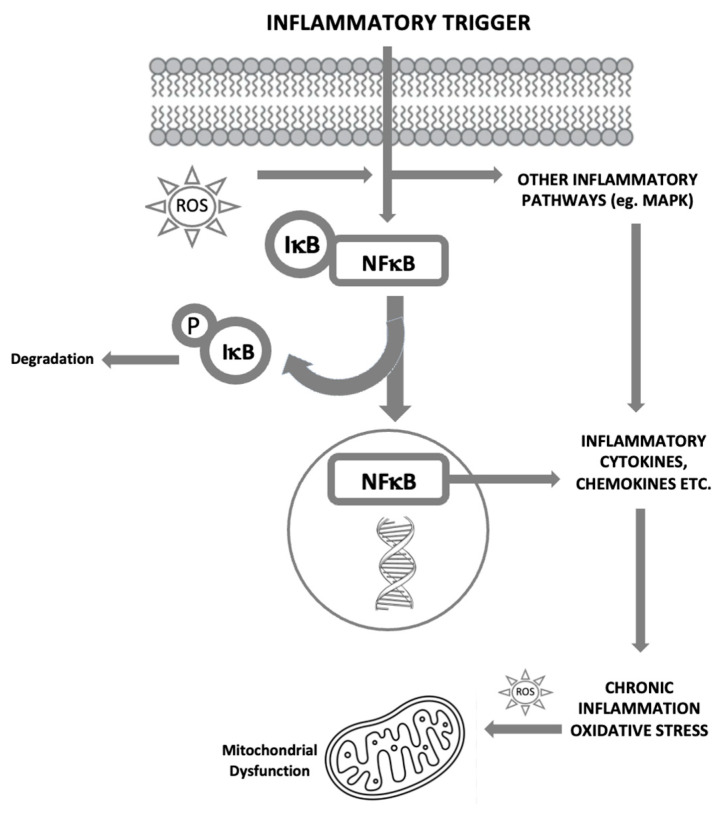
The bidirectional links between inflammation and oxidative stress. Reactive oxygen species (ROS) can act as inflammatory trigger initiating inflammation. On the other hand, inflammation induces oxidative stress. IkB, inhibitory subunit of NFkB; MAPK, mitogen-activated protein kinase; NFkB, nuclear factor kappa-light-chain-enhancer of activated B cells; P, phosphate; ROS, reactive oxygen species [19].

**Figure 3 ijms-24-10717-f003:**
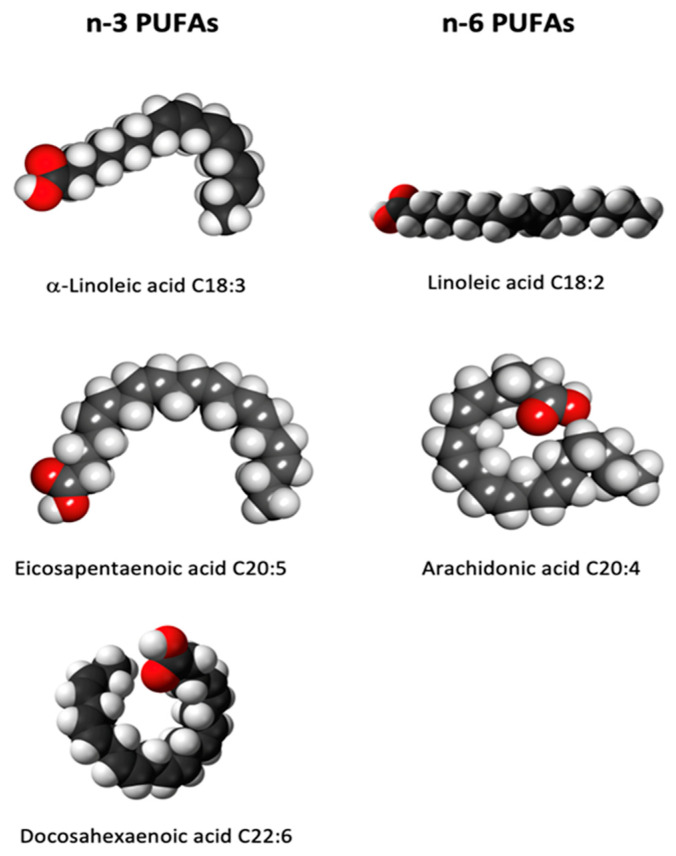
Space filling chemical structures of omega-3 and omega-6 PUFAs.

**Figure 4 ijms-24-10717-f004:**
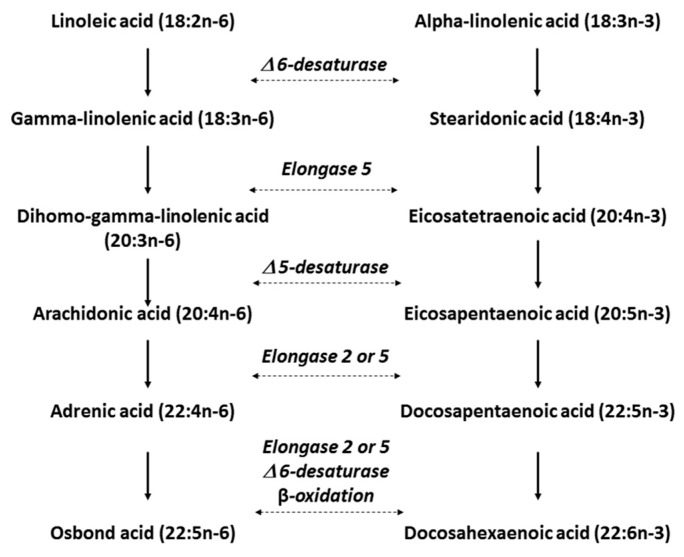
Pathway of metabolic interconversion of omega-6 and omega-3 polyunsaturated fatty acids. LA and ALA are the parent PUFAs for omega-6 and omega-3, respectively. Abbreviation used: Δ, delta [22].

**Table 1 ijms-24-10717-t001:** Commonly used scales for assessing PD.

**Unified Parkinson’s Disease Rating Scale (UPDRS):** The UPDRS is one of the most widely used scales for assessing Parkinson’s disease. It is a comprehensive scale that includes multiple sections to assess different aspects of motor function, including tremor, rigidity, bradykinesia (slowness of movement), and postural instability. The UPDRS also includes sections to assess non-motor symptoms, such as mood, cognition, and activities of daily living (ADLs).
**Hoehn and Yahr Scale:** The Hoehn and Yahr Scale is a widely used clinical staging scale that assesses the severity of Parkinson’s disease based on the presence and severity of motor symptoms. It ranges from stage 1 (mildest) to stage 5 (most severe) and is often used to classify the overall disease severity and track disease progression over time.
**Movement Disorder Society-Unified Parkinson’s Disease Rating Scale (MDS-UPDRS):** The MDS-UPDRS is a modified version of the UPDRS that has been developed by the Movement Disorder Society (MDS) to provide a more standardized and comprehensive assessment of Parkinson’s disease. It includes additional items and scoring criteria compared to the original UPDRS, and it has come to be widely used in research and clinical trials.
**Parkinson’s Disease Questionnaire (PDQ-39):** The PDQ-39 is a self-reported questionnaire that assesses health-related quality of life in individuals with Parkinson’s disease. It includes multiple domains, such as mobility, activities of daily living, emotional well-being, and social support, and provides a subjective assessment of the impact of Parkinson’s disease on the individual’s quality of life.
**Non-Motor Symptoms Scale (NMSS):** The NMSS is a scale specifically designed to assess non-motor symptoms of Parkinson’s disease, including mood disturbances, cognitive impairment, sleep disturbances, autonomic dysfunction, and sensory symptoms. It provides a comprehensive assessment of non-motor symptoms, which are common in Parkinson’s disease but may not be captured via traditional motor assessment scales.

## Data Availability

Not applicable.

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
