# Peer review of "DHA/EPA (Omega-3) and LA/GLA (Omega-6) as Bioactive Molecules in Neurodegenerative Diseases"

_ijms, 2023, doi:10.3390/ijms241310717_

Round 1
Reviewer 1 Report
The paper is a
accurate and essential review on the topic.
I think that Discussion could consider both inputs from Metabolomics and Individualized Therapies
Author Response
Thank you for the kind comment and now a paragraph has been inserted from lines 871 to 884 according to your instructions
Reviewer 2 Report
This review covers the roles of PUFAs and potential for therapeutic applications omega 3 PUFAs in neurodegenerative diseases. This review is on a relevant topic, likely to be of interest to readers. However, the manuscript could use some improvement before consideration for publication.
Abstract - The abstract should be shortened significantly in order to be in line with the journal guidelines (200 words maximum). It would also read better if it was more concise and to the point.
Main body - There are multiple statements made without a reference in this section. This should be fixed by adding reference citations to to all statements/sentences regarding what is known from research. It is not clear when there is one citation at the end of a paragraph, if that applies to all statements made within that paragraph.
There are a few places in the review where the language becomes informal or exaggerative words which do not belong in scientific writing are used. For example, line 94-95 "It is worth mentioning that cooking methods can seriously affect the fatty acid content" and line 118 "The human body can produce extraordinarily little amounts of linoleic acid". It is recommended that the authors carefully review and edit the manuscript to be more in line with a scientific writing style. Additionally, many sections would benefit from a thorough read through and editing to make them more concise and to the point rather than listing many details of studies.
For each of the disease sections, adding a table summarizing the key points of the studies discussed would be helpful to readers to organize the information.
On lines 669-672 the authors cite two different studies with the same reference. It appears that the mistake is at line 672 (reference 100 should be cited, not 99).
Conflict of interest statement - It is recommended that the authors reword the conflict of interest statement to first indicate IP's conflict of interest with the formula and company, then state that the remaining authors have no conflicts of interest. The current wording first makes it sound as if there are none, then describes the existing conflict of interest as an afterthought.
There are a few grammatical errors in the manuscript that should be addressed.
Author Response
Thank you for your kind comments that surely will make our manuscript read better. The abstract is now shortened to 200 words as instructed
References have now been added as precisely as it can possibly be and all have been double checked for relevance
As asked the lines 94-95 "It is worth mentioning that cooking methods can seriously affect the fatty acid content" and line 118 "The human body can produce extraordinarily little amounts of linoleic acid". Have been amended as follows: : "Cooking methods can affect the fatty acid content of various foods to a degree" and "The human body can produce limited amounts of linoleic (LA) and a-linoleic acid (ALA)"
As far as on your comment " It is recommended that the authors carefully review and edit the manuscript to be more in line with a scientific writing style. Additionally, many sections would benefit from a thorough read through and editing to make them more concise and to the point rather than listing many details of studies. "
We thank you on the productive comment and we have changed accordingly the document but we are kindly indicate that it is our believe that it is important to provide some details so that the reader has more clarity.
On your comment for possible tables for each of the discussed diseases, we agree but on the other hand we note that the parameters in the reviewed studies are much heterogeneous and the tables in such case would look more scattered and confusing than informative
For the comment "On lines 669-672 the authors cite two different studies with the same reference" is now amended appropriately
The comment on the conflict of interest has now been changed according to your instructions
Reviewer 3 Report
This review provides an overview of the existing recent clinical studies regarding the role of omega-3 and omega-6 PUFAs as therapeutic agents in chronic, inflammatory, autoimmune neurodegenerative diseases.
The authors introduced the most current neurodegenerative diseases classified as autoimmune diseases. Sections 2 and 3 explain the leading role of lipids.
Suggestions for improvement:
Figures 1 and 4 are not impressive and seem unnecessary.
Figure 2 has already been seen in the same journal (IJMS), and Figure 3 has already been reproduced in other MDPI journals. It would be advisable to create original figures.
In lines 158-160, the authors mentioned that „Molecules such as resolvins and protectins are nowadays identified as omega-3 PUFAs precursors and therefore contribute to the resolution of inflammatory processes in model systems. “ Resolvins and protectins are not omega-3 PUFAs precursors! Clarify this!
In lines 199-201, for the statement „Both human and animal studies have proved that diets high in DHA and EPA can increase the proportion of these PUFAs in the membranes of inflammatory cells and also reduce the levels of AA, “ references are missing.
In Section 4, references 39 and 40 are about PUFAs and PD, but it remains unclear which PUFAs were considered. Explain this! Reference 41 does not mention any PUFAs?
In lines 218-219, for the statement „More specifically, muscle and adipose tissue of meat are rich in ALA, EPA, DPA and DHA“, references are needed.
Regarding multiple sclerosis and reference 57, what changes were shown after a 3- month wash out?
The authors should consider technical elements (e.g., repeated abbreviations MUFAs and PUFAs, punctations, etc.).
This review provides valuable details regarding omega-3 and omega-6 PUFAs and neurodegenerative diseases.
The paper complies with the special issue of the journal.
Moderate editing of English language required.
Author Response
Dear Reviewer
thank you very much for your time to read and productively comment on our manuscript and we are now completed all indicated changes and amendments as follows:
Figure 1 has been removed but we are kindly believe that Figure 4 has to remain as it proved structural clarity
Figures 2 and 3 are now been redrawn
The sentence "Molecules such as resolvins and protectins are nowadays identified as omega-3 PUFAs precursors and therefore contribute to the resolution of inflammatory processes in model systems. “ has now been changed to: "Molecules such as resolvins and protectins are nowadays identified as molecules that are generated from omega-3 PUFA precursors and can orchestrate the timely resolution of inflammation in model systems"
As far as the comment for inserting reference for "both human and animal studies....." it is now been corrected
For the comment "...but it remains unclear which PUFAs were considered.." we are reporting that the specific studies are referring to PUFAs in general without indicating specific molecules and thus we keep the same format when are referencing on the indicative scientific reports...
Reference 41 is included because we strongly believe that PUFA as natural products and part of a common diet can relate to the concept of the reference 41. Even more, the reference 41 is based on a huge study with a very large number of participants which still playing a role as a base for future studies,
For a reference to be added ".....More specifically, muscle and adipose tissue.." it has been inserted
For "...Regarding multiple sclerosis and reference 57,..." the additional washout info has been inserted
The comment for "technical elements" has been addressed
Thank you for your kind comments
Round 2
Reviewer 2 Report
Overall the review article is improved and just about read for publication. It is still suggested that the authors proofread for grammatical errors. There were a few specific issues I noticed and have indicated below:
Abstract, lines 18-20: "The role of linoleic acid (LA) and gamma linolenic Acid (GLA) polyunsaturated fatty acids (omega-6 PUFAs) on neuroprotection is controversial as some of these agents, and specifically arachidonic Acid (AA),..." It is suggested that the authors revise this sentence. They way it currently reads makes it sound like AA is a subset of LA and GLA, rather than them all being omega 6 PUFAs. Perhaps something like the following would read better: "The role of omega-6 polyunsaturated fatty acids, such as linoleic acid, gamma linolenic acid, and arachidonic acid, on neuroprotection is controversial..."
Be sure to define abbreviations the first time they are used in the body of the manuscript (even if defined in the abstract). DHA/EPA and LA/GLA are mentioned on line 72 for the first time in the body, with no definition.
Lines 166-167: "EPA and ALA both compete for the same (Figure 4)." It appears that the authors left out part of this sentence.
Minor grammatical errors were found throughout. A thorough proofreading is recommended.
Author Response
Thank you for the comments
-lines 18-20 in abstract are now amended as suggested
-abbreviations are defined when found for the first time in the text
-lines 166-167 are now completed and a thorough proofreading has been performed
thank you